# MDACl_2_-Modified SnO_2_ Film for Efficient Planar Perovskite Solar Cells

**DOI:** 10.3390/molecules28062668

**Published:** 2023-03-15

**Authors:** Yaodong Xiao, Xiangqian Cui, Boyuan Xiang, Yanping Chen, Chaoyue Zhao, Lihong Wang, Chuqun Yang, Guangye Zhang, Chen Xie, Yulai Han, Mingxia Qiu, Shunpu Li, Peng You

**Affiliations:** College of New Materials and New Energies, Shenzhen Technology University, Shenzhen 518118, China

**Keywords:** perovskite solar cells, SnO_2_ modification, MDACl_2_, defect passivation

## Abstract

The electron transport layer (ETL) with excellent charge extraction and transport ability is one of the key components of high-performance perovskite solar cells (PSCs). SnO_2_ has been considered as a more promising ETL for the future commercialization of PSCs due to its excellent photoelectric properties and easy processing. Herein, we propose a facile and effective ETL modification strategy based on the incorporation of methylenediammonium dichloride (MDACl_2_) into the SnO_2_ precursor colloidal solution. The effects of MDACl_2_ incorporation on charge transport, defect passivation, perovskite crystallization, and PSC performance are systematically investigated. First, the surface defects of the SnO_2_ film are effectively passivated, resulting in the increased conductivity of the SnO_2_ film, which is conducive to electron extraction and transport. Second, the MDACl_2_ modification contributes to the formation of high-quality perovskite films with improved crystallinity and reduced defect density. Furthermore, a more suitable energy level alignment is achieved at the ETL/perovskite interface, which facilitates the charge transport due to the lower energy barrier. Consequently, the MDACl_2_-modified PSCs exhibit a champion efficiency of 22.30% compared with 19.62% of the control device, and the device stability is also significantly improved.

## 1. Introduction

In recent years, organic metal halide perovskite solar cells (PSCs) have shown great commercial prospects because of their high efficiency and low-cost processing [1,2,3,4,5,6]. Perovskite materials have unique optoelectronic properties, such as long carrier diffusion length and mobility, high light absorption coefficient, small exciton binding energy, and adjustable bandgap [7,8,9,10,11]. Therefore, perovskite materials have significant advantages as an active layer material of solar cells, compared with other traditional light-absorbing materials, such as silicon and GaN [12,13,14,15,16]. Since 2009, the power conversion efficiencies (PCEs) of PSCs have increased greatly from 3.8% to 25.7% [17,18,19,20,21]. The fast development of PSCs is inseparable from the device structure improvement, especially the optimization of the electron transport layer (ETL) [22,23,24]. ETLs in PSCs not only work as electron transport and hole-blocking layers but also their morphologies, structures, and surface properties will also affect the crystal growth conditions of the adjacent perovskite layer [25,26]. Therefore, ETL modification has become an effective way to further improve the performance of PSCs [27,28].

Titanium dioxide (TiO_2_) and Tin oxide (SnO_2_) films are widely used ETLs in high-efficiency PSCs. Both TiO_2_ and SnO_2_ have their distinctive advantages as ETLs, and device efficiencies have exceeded 20% in recent studies, thus it is worth considering which material is more appropriate [29,30,31,32]. Compared with the traditional TiO_2_ ETL, SnO_2_ has outstanding advantages, such as high electron mobility, suitable energy level alignment, good chemical stability, and a simple preparation process [33,34]. SnO_2_ films can be prepared at low temperatures, which is compatible with the preparation processes of flexible optoelectronic devices [35,36]. Therefore, the SnO_2_ ETL is considered more suitable for the future commercialization of PSCs. However, the SnO_2_ films tend to have surface or bulk defects caused by oxygen vacancies [37], which is not conducive to carrier transport and collection. Recent research has indicated that SnO_2_ modification by physical or chemical methods is an effective way to achieve simultaneous improvements in SnO_2_ ETL, perovskite layer, and the ETL/perovskite interface, which could significantly enhance the device performance [38,39]. Generally, the SnO_2_ modification strategies can be divided into two categories, surface modification, and bulk blending. The surface modification involves directly coating the modified material on the SnO_2_ surface as an interfacial buffer layer between the SnO_2_ and the perovskite film, while bulk blending indicates that the modified material is incorporated into the SnO_2_ precursor solution and is blended with the SnO_2_ to form a hybridized film [40,41,42]. Until now, various metal cations such as K^+^, Mg^2+^, Al^3+^, and Ga^3+^ have been utilized for the SnO_2_ bulk blend modification, and the incorporated external elements can modulate the SnO_2_ self-doping defects, resulting in enhanced n-type conduction of the films [35,43,44]. In addition, the halide ions (F^−^, Cl^−^, and I^−^) have also been widely used for the bulk blending modification of the SnO_2_ films. This strategy based on halide incorporation improves the SnO_2_ film absorbance and reduces the defect concentration in the SnO_2_/perovskite interface region [37,45,46,47]. For example, in 2019, Liu et al. dispersed NH_4_Cl in SnO_2_ colloidal solution for the first time [48]. The introduced NH_4_^+^ and Cl^−^ ions help enhance the charge transport capability of the ETL and inhibit the non-radiative recombination at the ETL/perovskite interface. As a result, the efficiency of the planar PSCs based on NH_4_Cl-doped SnO_2_ ETL was improved to 21.38%, compared to the 18.71% efficiency of the control device. In 2021, Lin et al. reported a simple and effective method for electron transport layer modification by introducing methylamine hydrochloride (MACl) and formamidine hydrochloride (FACl) into the SnO_2_ colloidal solution [49]. These dopants promote the nucleation and growth of the perovskite crystals on the ETL, leading to the improved quality of the perovskite films. Meanwhile, the energy level arrangement between the ETL and perovskite layers is adjusted, and the carrier trap density is reduced. Compared with the undoped devices, the efficiency of MACl/FACl-modified devices increased to 21.87% and 21.72%, respectively, which proves that the ETL precursor solution engineering strategy is an effective way to obtain high-performance PSCs. The bulk blending modification strategy modulated SnO_2_ properties while passivating the SnO_2_/perovskite interfacial defects, therefore we attempted to find a novel material for SnO_2_ modification. Methylenediammonium dichloride (MDACl_2_) was reported to have a stabilizing effect on the α-phase FAPbI_3_, and the corresponding devices were prepared with PCEs reaching 23.7% and 25.5% [50,51]. Although MDACl_2_ has been demonstrated to be significantly effective in optimizing the performance of perovskite layers, their applications in SnO_2_ ETL have never been reported previous in the literature.

In this study, we introduced a type of small molecule material MDAC1_2_ into the SnO_2_ precursor solution for the first time and systematically explored the effect of MDAC1_2_ incorporation on the SnO_2_ film and PSCs. The SnO_2_ film modification strategy not only passivates surface defects on the film surface but also reduces the energy barrier at the SnO_2_/perovskite interface. Meanwhile, the improved SnO_2_ surface property is also beneficial for the nucleation and growth of the perovskite grains, resulting in enhanced crystallinity of the perovskite films, which reduces the carrier defect density and suppresses the non-radiative recombination. Consequently, the PCE of the MDAC1_2_-modified device was increased to 22.30% compared with the PCE of 19.62% for the control device. This study demonstrates the effectiveness of the ETL precursor solution modification strategy for performance enhancement of the PSCs.

## 2. Results and Discussion

### 2.1. Device Performance Distribution

A dense electron transport layer was deposited on the FTO substrate based on the SnO_2_ colloidal precursor solution containing different concentrations of MDACl_2_. The corresponding PSCs with the device structure shown in Figure 1a were prepared. The distributions of photovoltaic parameters including open-circuit voltage (V_OC_), short-circuit current density (J_SC_), fill factor (FF), and PCE of the PSCs prepared under conditions of different MDACl_2_ concentrations are shown in Figure 1b–e. The statistic photovoltaic parameters are listed in Table 1. The PSCs with undoped SnO_2_ as ETL (control) show an average PCE of 18.92%, V_OC_ of 1.075 V, J_SC_ of 23.35 mA/cm^2^, and FF of 75.39%. In contrast, the PSCs with MDACl_2_-modified SnO_2_ as ETL show enhanced device performance than that of the control devices. The best device performance was achieved when 4 mM MDACl_2_ was added to the SnO_2_ precursor solution, with an average PCE of 21.53%, V_OC_ of 1.105 V, J_SC_ of 24.45 mA/cm^2^, and FF of 79.65%. The underlying working mechanism of the MDACl_2_ additive is discussed in the following part.

We have compared the performances of PSCs prepared by using MDACl_2_-incorporated SnO_2_ precursor solution and direct spin-coating MDACl_2_ solution (4 mM) on top of the SnO_2_ film, as shown in Appendix A. Notably, the introduction of MDACl_2_ molecules in both conditions all contribute to improved device performances. However, the average PCE of PSCs prepared based on MDACl_2_-incorporated SnO_2_ precursor solution (average PCE: 20.41%) is much lower than that of the devices prepared by direct spin-coating MDACl_2_ buffer layer on top of the SnO_2_ film (average PCE: 21.53%). Therefore, we believe that the incorporated MDACl_2_ molecules not only locate on the SnO_2_ film surface but also modify the interconnection parts of the SnO_2_ nanoparticles in the SnO_2_ thin film [52]. Furthermore, the concentration of MDACl_2_ we introduced into the SnO_2_ solution here was very low (4 mM), and it was insufficient to form a continuous MDACl_2_ film on top of the SnO_2_ film, which can be confirmed by the AFM images shown in Appendix A. To better understand the reasons for the efficiency enhancement of the MDACl_2_-modified PSCs, we further investigated the ETLs for more details.

### 2.2. ETL Characterizations

Based on the device efficiency distributions, we further compared the difference between the unmodified SnO_2_ and MDACl_2_-modified SnO_2_ films (4 mM). For convenience, we define them as SnO_2_ and SnO_2_-MDACl_2_ films, respectively. The surface morphologies of the ETL films were characterized by an atomic force microscope (AFM). As shown in Appendix A, the roughness of the two films is quite similar, and the average roughness of the SnO_2_-MDACl_2_ film (8.1 nm) is slightly lower than that of the control SnO_2_ film (8.8 nm). The transmittance spectra of the ETL films (see Appendix A) show that MDACl_2_ modification has little effect on the optical transmittance of the SnO_2_ film. The effect of MDACl_2_ modification on the energy band arrangement of the ETLs was evaluated by ultraviolet photoelectron spectroscopy (UPS) and UV-visible absorption spectra (UV-vis). As shown in Figure 2a,b, the Fermi levels (E_F_) of the SnO_2_ and SnO_2_-MDACl_2_ films are −5.50 eV and −5.66 eV, respectively, and the valence band maximum (VBM) are −8.29 eV and −8.13 eV, respectively [49]. According to the absorption spectra of the ETL films (see Appendix A), the band gaps (E_g_) of the SnO_2_ and SnO_2_-MDACl_2_ films are both 3.90 eV. Therefore, the conduction band minimum (CBM) of the SnO_2_ and SnO_2_-MDACl_2_ films can be calculated to be −4.39 eV and −4.23 eV (Appendix A), respectively [53,54]. The energy level arrangement of the two ETL films and the perovskite film is shown in Figure 2c. Compared with the SnO_2_ film, the CBM of the SnO_2_-MDACl_2_ film is closer to that of the perovskite film, which is more conducive to the transport and extraction of photogenerated charges at the ETL/perovskite interface.

The incorporation effects of MDACl_2_ in the ETLs were further evaluated by X-ray photoelectron spectroscopy (XPS). The XPS full spectra shown in Appendix A confirm the existence of Sn and O elements in the SnO_2_ and SnO_2_-MDACl_2_ films. As shown in the high-resolution Cl 2p spectra (Figure 2d), a characteristic peak (198.3 eV) was detected in the SnO_2_-MDACl_2_ film, while the peak was not found in the SnO_2_ control film, indicating that the MDACl_2_ material was successfully incorporated in the SnO_2_-MDACl_2_ film. As shown in Figure 2e, the Sn 3d characteristic peak (486.6 eV) of the SnO_2_-MDACl_2_ film is shifted by 0.2 eV to the high binding energy direction compared to the peak (486.4 eV) of the control SnO_2_ film [55], indicating that the chemical environment of the Sn element has changed. Appendix A demonstrates the C 1s peaks, and the C−C/C−H components are positioned at 284.8 eV. In addition, the C−O and O−C=O bonds could be attributed to external adventitious carbon contamination and SnO_2_ surface defects [56,57]. From Appendix A, the O 1s characteristic peak splits into two parts corresponding to O^2−^ and OH^−^, and the proportion of OH^−^ is 34.88% for the SnO_2_ film and 31.35% for the SnO_2_-MDACl_2_ film [58,59], respectively. The slightly reduced OH^−^ proportion can be attributed to the incorporation of MDACl_2_ molecules in the SnO_2_ film. In addition, the K 2p peak is observed at 292.7 eV (Appendix A), which is attributed to the presence of K^+^ (as a stabilizer) in the commercial SnO_2_ colloidal solution. Therefore, the incorporation of MDACl_2_ as an additive in the SnO_2_ precursor solution was demonstrated to be an effective strategy for enhancing film conductivity and passivating surface defects of the SnO_2_ films. The modification of the surface, as well as the inner part of the SnO_2_ film by MDACl_2_ molecules, contribute together to the improved film properties of the SnO_2_ film.

As shown in Appendix A, the contact angles of water droplets on the SnO_2_ and SnO_2_-MDACl_2_ substrates were measured to be 41.25° and 46.90°, respectively. The higher contact angle of the SnO_2_-MDACl_2_ film is consistent with the decreased concentration of the hydroxyl groups on the SnO_2_-MDACl_2_ film surface. The wettability of the substrate after MDACl_2_ treatment is lower, which is beneficial to inhibit the heterogeneous nucleation during the following perovskite film deposition process and reduce the number of nucleation centers [61,62,63,64]. Meanwhile, the increase in contact angle indicates a decrease in surface energy, which reduces the energy barrier for crystal growth and promotes the vertical growth of the perovskite grains. Therefore, the MDACl_2_ modification of the ETLs will contribute to the formation of high-quality perovskite films, which will be further discussed in the following part.

The electrical conductivity changes of the ETL films can be obtained by measuring the current-voltage (I–V) characteristics of the FTO/SnO_2_/Ag devices, as shown in Figure 2f. The conductivity can be calculated according to Formula (1) [65,66]:(1)$$I=\sigma_{0}Ad^{-1}V$$
where *I* is current, *σ*_0_ is the ETL film conductivity, *A* is the active area, *d* is the thickness of the ETL film, and *V* is the voltage. Regarding the SnO_2_ film thickness, we obtained the cross-sectional SEM image of the FTO/SnO_2_-MDACl_2_/Perovskite sample (Appendix A) and estimated the SnO_2_ film thickness to be around 40 nm. The calculated conductivity results are shown in Appendix A. Incorporating MDACl_2_ into the SnO_2_ ETL can enhance the film conductivity at different levels. When the MDAC1_2_ concentration is 4 mM, the maximum electrical conductivity of the ETL is calculated to be 5.27 × 10^−3^ mS/cm, which is 1.93 times that of the control film (2.73 × 10^−3^ mS/cm). The increased ETL conductivity can accelerate electron extraction and transportation.

### 2.3. Perovskite Film Characterizations

The surface morphologies of the perovskite films prepared on the SnO_2_ and SnO_2_-MDACl_2_ films were characterized by scanning electron microscopy (SEM), as shown in Figure 3a,b. Notably, the average grain sizes of the perovskite films grown on SnO_2_ and SnO_2_-MDACl_2_ ETLs are around 700 nm and 1000 nm, respectively. The larger perovskite grain sizes represent a decrease in grain boundary concentration, thereby reducing the non-radiative carrier recombination caused by grain boundary defects [67], which is beneficial to carrier transport. Then we further carried out XRD characterizations of the perovskite films prepared on the SnO_2_ and SnO_2_-MDACl_2_ films, as shown in Figure 3c. Both perovskite films have two strong diffraction peaks, corresponding to the (110) and (220) lattice planes, respectively. The full width at half maximum (FWHM) values of the perovskite (110) XRD peaks based on the SnO_2_ and SnO_2_-MDACl_2_ films are 0.086 and 0.077 (degree), respectively. The smaller FWHM value of the SnO_2_-MDACl_2_/perovskite film indicates the increased average perovskite grain size, which is consistent with the SEM results. Notably, the diffraction peaks of the perovskite film on SnO_2_-MDACl_2_ ETL are stronger, which indicates that the perovskite crystallinity is improved [68]. Furthermore, the UV-Visible absorption spectra of the perovskite films prepared on the SnO_2_ and SnO_2_-MDACl_2_ films were obtained (see Figure 3d). In the wavelength range of 500–750 nm, the absorption intensity of the perovskite film prepared on the SnO_2_-MDACl_2_ ETL is significantly higher than that of the control film, indicating the enhanced light absorption properties of the SnO_2_-MDACl_2_/perovskite films. Moreover, the energy band edge of the perovskite film calculated from the Tauc plots (inset of Figure 3d) were both 1.56 eV, which means that the perovskite film prepared on the SnO_2_-MDACl_2_ ETL has a comparable band gap with that of the control film [69]. Therefore, the MDACl_2_ modification contributes to the formation of high-quality perovskite films with increased crystallinity and light absorption properties.

Steady-state photoluminescence (SSPL) and time-resolved photoluminescence (TRPL) measurements were utilized to further study the charge dynamics at the ETL/perovskite interface. As shown in Figure 4a, the luminescence intensity of the SnO_2_-MDACl_2_/perovskite film is significantly reduced compared to the undoped SnO_2_ ETL, indicating that MDACl_2_ incorporating is beneficial to charge extraction and transport at the SnO_2_/perovskite interface [37,70,71]. This should be ascribed to the increased conductivity and the passivated surface defects of the SnO_2_ film. This conclusion is further verified by the TRPL characterization results of the ETL/perovskite films (Figure 4b). Notably, the PL decay rate of the SnO_2_-MDACl_2_/perovskite film is faster than that of the control sample. The decay time constants were calculated by using the double exponential function fitting, and the results are summarized in Appendix A. The average decay time constants (τ_ave_) of the SnO_2_/perovskite and SnO_2_-MDACl_2_/perovskite are 91.27 ns and 55.64 ns, respectively, indicating that the SnO_2_-MDACl_2_ ETL has higher electron extraction ability [52,72]. The stronger PL quenching implies enhanced ETL charge extraction, whereas the simultaneously enhanced non-radiative recombination and lower quasi-Fermi energy level splitting are detrimental to the device V_OC_ [73,74,75,76]. Therefore, further analysis is required for the device charge transfer performance. Based on the space-charge-limited current (SCLC) technique, we fabricated the electronic-only devices and evaluated the defect state density of the perovskite films prepared on the SnO_2_ and SnO_2_-MDACl_2_ ETLs. Figure 4c,d illustrates the dark I–V curves of the electron-only devices with a device structure of FTO/ETL/Perovskite/PCBM/Ag. The trap-filling limit voltage (*V_TFL_*) can be obtained by fitting the dark I–V curve. The *V_TFL_* value of the electron-only device based on SnO_2_-MDACl_2_ ETL (0.308 V) is lower than that of the SnO_2_ ETL (0.438 V). The trap densities of the perovskite films can be calculated according to Formula (2) [44,77,78]:(2)$$n_{trap}=\frac{2\varepsilon \varepsilon_{0}V_{TFL}}{eL^{2}}\;$$
where *ε* represents the dielectric constant of perovskite (*ε* = 31.18), *ε*_0_ is the vacuum permittivity, *e* is the elementary charge, and *L* is the thickness of the perovskite film (*L* = 550 nm). After calculation, the trap density of the perovskite film decreased from 4.99 × 10^15^ cm^−3^ to 3.51 × 10^15^ cm^−3^ after MDACl_2_ modification, which is attributed to the improved perovskite crystallinity [58,63].

### 2.4. Device Characterizations

Figure 5a shows the J–V curves and photovoltaic parameters of the best-performing PSCs prepared based on the unmodified SnO_2_ (control) and SnO_2_-MDACl_2_ ETLs. The best control device shows a champion efficiency of 19.62%, with V_OC_ of 1.077 V, FF of 77.06%, and J_SC_ of 23.64 mA/cm^2^, while the PSCs prepared based on the SnO_2_-MDACl_2_ ETL demonstrate a champion efficiency of 22.30%, with V_OC_ of 1.111 V, FF of 81.40% and J_SC_ of 24.65 mA/cm^2^. The significant improvement of the device efficiency is mainly attributed to the enhancement of FF and V_OC_, which is mainly attributed to the film quality improvement of SnO_2_ and perovskite. Figure 5b shows the external quantum efficiency (EQE) and the integrated current density of the PSCs. The SnO_2_-MDACl_2_ device exhibits higher EQE values with an integrated current density of 24.06 mA/cm^2^, compared with that of the control device (23.11 mA/cm^2^), which is related to the enhanced charge extraction and transport ability of the MDACl_2_-based SnO_2_ film [79]. In addition, MDACl_2_ incorporation also reduces the J-V hysteresis of the PSCs (see Appendix A). The PSC prepared on the SnO_2_-MDACl_2_ ETL demonstrates high PCEs of 22.30% (reverse scan) and 22.28% (forward scan), respectively, showing almost no hysteresis, while PCEs of 19.62% (reverse scan) and 19.36% (forward scan) were obtained for the control device.

The stable power output measurements of the PSCs were performed at the maximum power points, as shown in Figure 5c. The PSC based on the SnO_2_-MDACl_2_ ETL achieves a stable PCE of 22.07%, while the control device only gets a 19.37% efficiency. Notably, the time to reach the stable PCE is much shorter for the target PSCs than that of the control device, meaning that the MDACl_2_-modified PSCs have smaller capacitive currents, which is consistent with the reduced J–V hysteresis of the PSCs [80,81]. To further evaluate the effects of MDACl_2_ modification on the long-term stability of the device, we compared the performance of the unencapsulated devices stored in a 35% humidity environment, as shown in Figure 5d. After aging for 600 h, the MDACl_2_-modified device maintains 93.1% of its initial efficiency, whereas the PCE of the control device decays to 81.3% of its initial efficiency. These results indicate that MDACl_2_ incorporation into the SnO_2_ ETL can greatly improve long-time device stability.

In addition, to better understand the reasons for the improved device performance after MDACl_2_ modification, we carried out light intensity dependence characterizations of the PSCs (see Figure 6a,b), to estimate the charge recombination kinetics inside the devices. Figure 6a shows the relationship between J_SC_ and light intensity. The PSCs based on the SnO_2_ and SnO_2_-MDACl_2_ ETLs exhibit very similar fitting slopes (α values), indicating that the devices have similar bimolecular recombination conditions [82,83]. Figure 6b shows the dependence of V_OC_ on light intensity. Notably, the derived ideal factor of the MDACl_2_-modified PSC (1.36) is smaller than that of the control device (1.53), which indicates the reduction of SRH monomolecular recombination in the MDACl_2_-modified PSC [48,84].

The dark-state J–V curves of the PSCs are shown in Figure 6c. The decreased leakage current of the MDACl_2_-modified PSC indicates the reduced defect density and defect-assisted non-radiative recombination after MDACl_2_ modification [85]. We further evaluated the charge transport performance by electrochemical impedance spectroscopy (EIS) measurements in dark conditions at a bias voltage of 0.8 V. Figure 6d shows the Nyquist plots, from which the series resistance (R_s_) and recombination resistance (R_rec_) are derived, as listed in Appendix A. The Rs value is decreased from 37.16 Ω (control device) to 30.78 Ω (MDACl_2_-modified PSC), confirming the improved charge transport in the optimized PSC. Moreover, the MDACl_2_-modified PSC demonstrates an increased R_rec_ (337.3 Ω) compared to that of the control device (217.3 Ω), which indicates the suppressed charge recombination inside the MDACl_2_-modified PSC [24,86].

## 3. Materials and Methods

### 3.1. Materials

Formamidinium iodide (FAI), methylammonium bromide (MABr), methylammonium chloride (MACl), and Butylammonium Iodide (BAI) were ordered from Greatcell Solar Ltd. PbI_2_ (99.999%) and SnO_2_ (15 wt % colloidal dispersion tin (IV) oxide) was purchased from Alfa Aesar. Cesium iodide (CsI) (99.999%), dimethylformamide (DMF), dimethylsulfoxide (DMSO), isopropanol, chlorobenzene (CB), ethyl acetate, 4-tert-Butylpyridine (tBP), bis(trifluoromethylsulphonyl)imide lithium salt (Li-TFSI), acetonitrile, and methylenediamine dihydrochloride (MDACl_2_) were all bought from Sigma-Aldrich Ltd. (Saint Louis, MO, USA) PbBr_2_ (99.999%) was purchased from Xi’an Polymer Light Technology Corp. (Xi’an, China) Spiro-OMeTAD was purchased from Lumtec Ltd. (Taiwan, China) All the chemicals and reagents were used as received without further purification.

### 3.2. Device Fabrication

PSCs were fabricated through a one-step solution process with a formal structure of glass/FTO/SnO_2_/perovskite/Spiro-OMeTAD/Au. The FTO substrates were scrubbed with soap water and ultrasonically cleaned sequentially in deionized water, acetone, and isopropanol, before drying with nitrogen flow. Prior to use, the FTO substrates were treated with UV-ozone for 15 min. The commercial SnO_2_ colloid solution was diluted to 2.5 w% with deionized water and mixed with different concentrations of MDACl_2_. Subsequently, 80 μL mixed SnO_2_ solution was dropped on the FTO substrates and spin-coated at 4000 rpm for 30 s. After sintering at 100 °C for 30 min, the film was treated under UV-ozone for 15 min and transferred to the glove box immediately. The Cs_0.05_FA_0.89_MA_0.06_Pb(I_0.94_Br_0.06_)_3_ perovskite precursor solution was prepared by dissolving 726 mg PbI_2_, 258 mg FAI, 11.2 mg MABr, 36.7 mg PbBr_2_, 19.5 mg CsI and 35 mg MACl in 1 mL mixed solvent (DMF: DMSO = 8:1). The solution was stirred overnight before use. The 60 μL Cs_0.05_FA_0.89_MA_0.06_Pb(I_0.94_Br_0.06_)_3_ perovskite precursor solution was dropped on the SnO_2_ layer at 1000 rpm for 5 s and 5000 rpm for 25 s. At the 20th second after the start of the spin coating process, 300 μL ethyl acetate as antisolvent was dripped onto the perovskite film. Then the perovskite film was annealed at 150 °C for 10 min on a hot plate immediately. After cooling down to room temperature, 50 μL of BAI solution (1 mg mL^−1^ in IPA) was spin-coated onto the perovskite film followed by 100 °C annealing on the hotplate for 10 min. Subsequently, 75 mg of Spiro-OMeTAD was dissolved in 1 mL CB. Then the solution was doped with 28.5 μL tBP and 17.5 μL Li-TFSI (520 mg/mL in acetonitrile) to prepare the hole transporting layer (HTL) solution. Afterwards, 80 μL HTL solution was spin-coated on the perovskite films at 4000 rpm for 30 s. Finally, an 80 nm Au electrode was deposited on the HTL by thermally evaporating under the condition of 3 × 10^−5^ Pa, and the active area of all devices is 0.06 cm^2^ defined by a metal mask.

### 3.3. Characterizations

A field emission scanning electron microscope (GeminiSEM 300, Carl Zeiss Microscopy Ltd., Jena, Germany) was used to obtain the morphology of the perovskite films based on SnO_2_ and SnO_2_-MDACl_2_ ETLs. The surface morphology of SnO_2_ and SnO_2_-MDACl_2_ films was measured with an atomic force microscope (AFM) (Cypher S, Oxford Instruments Asylum Research, Inc., Oxford, UK). X-ray diffraction system (SmartLab XRD, Rigaku, Tokyo, Japan) was performed to characterize the crystallinity and crystal structure of Cs_0.05_FA_0.89_MA_0.06_Pb(I_0.94_Br_0.06_)_3_ perovskite films. The UV−Vis absorption spectra and transmission spectra were measured with a UV−Vis spectrophotometer (UV-2600, Shimadzu, Kyoto, Japan). Both the steady-state photoluminescence (PL) and time-resolved photoluminescence (TRPL) were performed on a steady-state transient fluorescence lifetime test system (FluoTime 300, PicoQuant, Berlin, Germany). The perovskite films for all PL measurements were prepared on FTO/SnO_2_ substrates and used a 405 nm laser as the excitation light source (illuminated from the perovskite film side). The Ultraviolet photoemission spectroscopy (UPS) and the X-ray photoemission spectroscopy (XPS) analysis of SnO_2_ and SnO_2_-MDACl_2_ films on glass substrates using an XPS Escalab Xi+ (Thermo Fisher Scientific (China) Co., Ltd., Shanghai, China). Curve fitting was performed using Thermo Avantage software. The XPS curve was calibrated for the C 1s peak at 284.8 eV. The contact angles of the samples were measured by the contact angle measuring instrument (Fed-A, Dongguan Furunde Intelligent Equipment Co., Ltd., Dongguan, China).

### 3.4. Device Measurements

The current density–voltage (J–V) curves of the PSCs were measured under standard 1-sun light illumination of 100 mW/cm^2^ (Enlitech solar simulator, equipped with an AM 1.5 filter). The light intensity was calibrated before the J–V test by using a standard reference silicon cell (Enlitech, Taiwan, China). The J–V measurements were conducted by forward scan (from 0 to 1.2 V) and reverse scan (from 1.2 to 0 V), with a scan rate of 20 mV/s. During the J–V measurement of devices, we used a metal aperture mask to ensure the reliability of the measurement. EQE was measured using an Enlitech QE-S EQE system (Taiwan, China) equipped with a standard Si detector and monochromatic light (Enlitech 300 W lamp source). The measurements were performed in AC mode at room temperature. The EQE response from a wavelength of 300 nm to 900 nm was recorded by a computer. The stabilized power output (SPO) was measured at the maximum power point of the PSCs under simulated AM 1.5G, 100 mW/cm^2^ solar irradiation by Enlitech solar simulator with an AM 1.5G filter. The Electrochemical Impedance Spectroscopy (EIS) measurements were performed by Modulab XM electrochemical workstation (Advanced Measurement Technology Inc., Oak Ridge, TN, USA).

## 4. Conclusions

In summary, we have proposed an effective and convenient precursor solution doping strategy for ETL modification. After MDACl_2_ modification, the surface defects of the SnO_2_ film are effectively passivated and the film conductivity is also improved, which is beneficial to the charge extraction and transport at the ETL/perovskite interface. Meanwhile, the MDACl_2_ modification contributes to the formation of high-quality perovskite films with high crystallinity and low defect density. Moreover, better energy level alignment is achieved at the ETL/perovskite interface, which can facilitate the charge transport due to the lower energy barrier. As a result, the champion PSC based on the MDACl_2_-modified SnO_2_ film demonstrates a PCE of 22.30%, with negligible hysteresis and enhanced device stability.

## Figures and Tables

**Figure 1 molecules-28-02668-f001:**
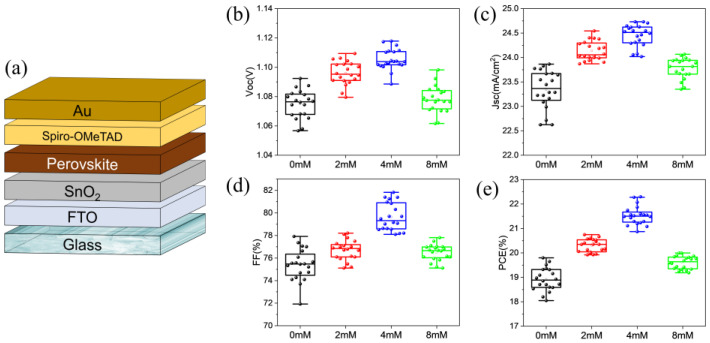
(**a**) Schematic diagram of the glass/FTO/SnO_2_/Perovskite/Spiro/Au PSC structure. (**b**) V_OC_, (**c**) J_SC_, (**d**) FF, and (**e**) PCE distributions of PSCs prepared under conditions of different MDACl_2_ concentrations. The data were collected from 20 devices for each condition.

**Figure 2 molecules-28-02668-f002:**
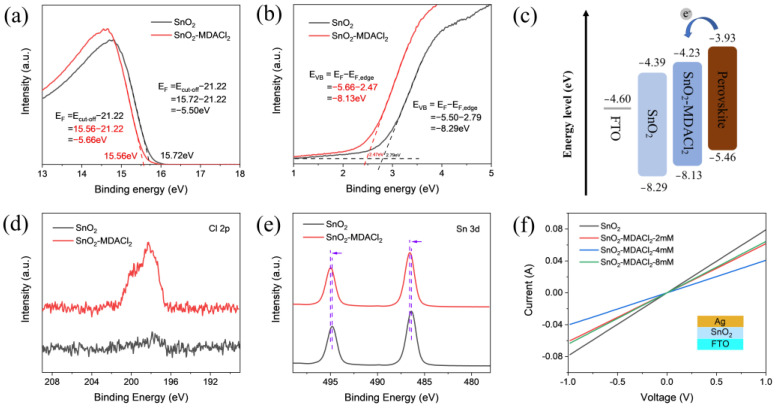
(**a**,**b**) UPS spectra of the SnO_2_, SnO_2_-MDACl_2_ films. (**c**) Energy band diagram of the SnO_2_, SnO_2_-MDACl_2_, and perovskite films. The ETL energy level alignments are based on the UPS and UV−vis measurements, and the perovskite energy level alignments are based on the literature listed [60]. (**d**,**e**) High-resolution Cl 2p and Sn 3d XPS spectra of the SnO_2_ and SnO_2_-MDACl_2_ films. (**f**) Conductivity measurement curves of the FTO/SnO_2_/Au sample with different MDACl_2_ doping concentrations.

**Figure 3 molecules-28-02668-f003:**
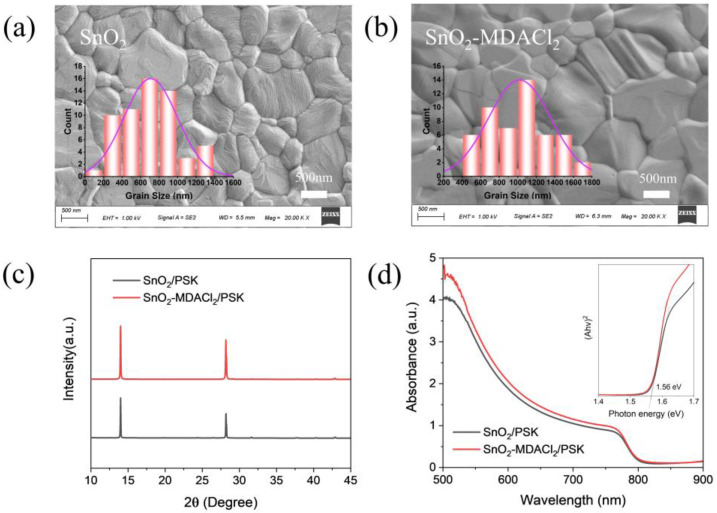
(**a**,**b**) SEM images of perovskite films grown on SnO_2_ and SnO_2_-MDACl_2_ ETLs. Insets show the statistical diagrams of the grain size distributions based on the SEM images. (**c**) XRD patterns of perovskite films grown on FTO/SnO_2_ and FTO/SnO_2_-MDACl_2_ substrates. (**d**) UV-visible absorption spectra and Tauc plots of the perovskite films grown on SnO_2_ and SnO_2_-MDACl_2_ ETLs.

**Figure 4 molecules-28-02668-f004:**
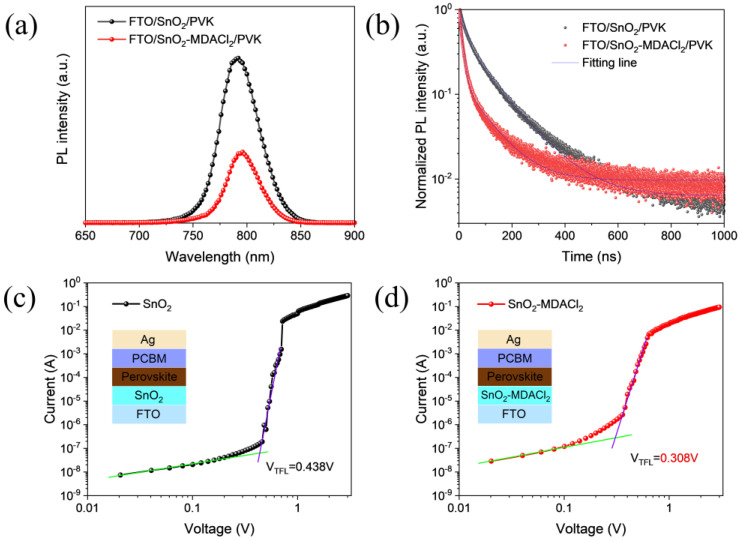
(**a**,**b**) Steady-state photoluminescence (SSPL) and time-resolved photoluminescence (TRPL) spectra of perovskite films deposited on the SnO_2_ and SnO_2_-MDACl_2_ ETLs. (**c**,**d**) Dark I–V curves of the electron-only devices. Insets show the schematic device structures.

**Figure 5 molecules-28-02668-f005:**
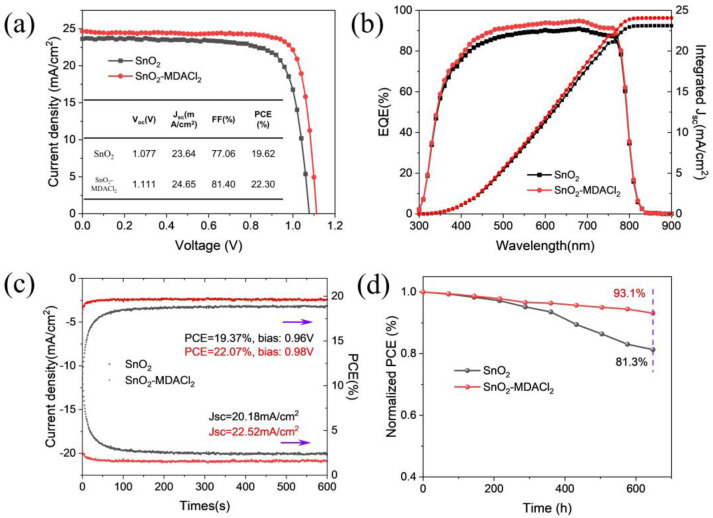
(**a**) J–V curves of the champion devices based on SnO_2_ and SnO_2_-MDACl_2_ substrates under reverse scan. (**b**) EQE spectra and the integrated current density of the PSCs. (**c**) Stable power output curves of the PSCs measured at the maximum power point under AM 1.5G illumination in glovebox. (**d**) Long-term stability performance of the unencapsulated PSCs under conditions of RH: 35%, T: 25 °C.

**Figure 6 molecules-28-02668-f006:**
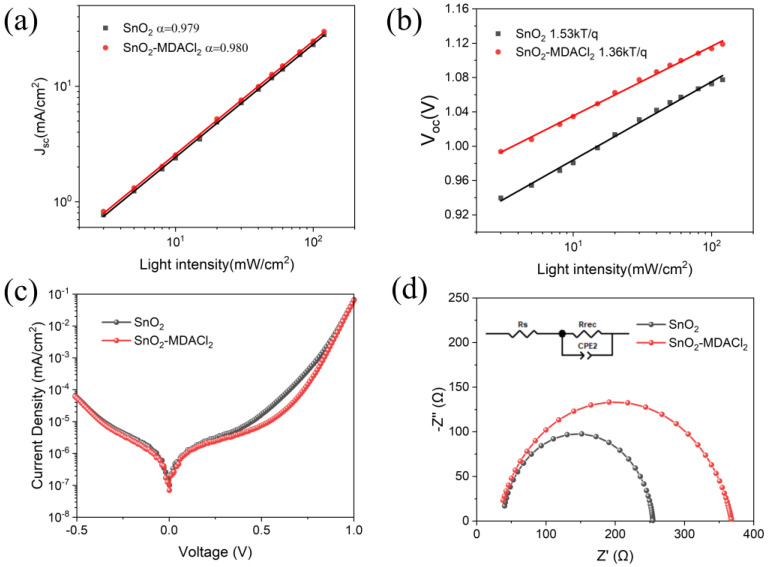
(**a**,**b**) Light intensity dependence characterizations of J_SC_ and V_OC_ for PSCs based on the SnO_2_ and SnO_2_-MDACl_2_ ETLs. (**c**) Dark-state J–V curves of the PSCs. (**d**) Nyquist plots for the PSCs (measured in dark conditions).

**Table 1 molecules-28-02668-t001:** Statistic photovoltaic parameters of PSCs prepared under conditions of different MDACl_2_ concentrations.

ETL	V_OC_ (V)	J_SC_ (mA/cm^2^)	FF (%)	PCE (%)
SnO_2_	1.075 ± 0.010	23.35 ± 0.39	75.39 ± 1.42	18.92 ± 0.47
SnO_2_-MDACl_2_-2 mM	1.096 ± 0.008	24.14 ± 0.20	76.71 ± 0.88	20.29 ± 0.27
SnO_2_-MDACl_2_-4 mM	**1.105 ± 0.007**	**24.45 ± 0.22**	**79.65 ± 1.24**	**21.53 ± 0.38**
SnO_2_-MDACl_2_-8 mM	1.078 ± 0.009	23.78 ± 0.21	76.48 ± 0.72	19.61 ± 0.26

## Data Availability

Not applicable.

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
