# Peer review of "MDACl2-Modified SnO2 Film for Efficient Planar Perovskite Solar Cells"

_molecules, 2023, doi:10.3390/molecules28062668_

Round 1

Reviewer 1 Report

In this manuscript, Xiao et al. propose an effective ETL modification strategy based on the incorporation of MDACl2 into the SnO2 precursor colloidal solution. The additive would not only passivate the surface defects of the SnO2 film but also contribute to the formation of high-16 quality perovskite films with improved crystallinity and reduced defect density. Consequently, the MDACl2-modified PSCs 19 exhibit a champion efficiency of 22.30% compared with 19.62% of the control device, and the device 20 stability is also significantly improved. I think the results are interesting and the manuscript can be accepted after addressing the following concerns.

1.      In the introduction part, the authors briefly summarize the modified SnO2 colloidal solution with two papers. As a lot of related work have been reported in this area with some impressive achievements, the authors should summarize them in a more comprehensive way to give readers a clearer picture on the promising of this topic. Meanwhile, the authors should emphasize the motivation and the advantages of employed MDACl2 additive compared to the reported similar additives, for example NH4Cl, MACl, and FACl.

2.      The authors conclude that the light absorption properties of perovskite films in the range of 500-750 nm were enhanced after introducing MDACl2 into SnO2 solution. The reason is explained by the formation of high-quality perovskite films with increased crystallinity. This is difficult to understand. More explanation and evidence are needed.

3.      Regarding the stability text, SnO2 show much fasted degradation after 200 h. But, at the initial 200 hours, SnO2 and SnO2-MDACl2 based devices exhibit similar degradation rates. This is strange and needs to be explained.

Reviewer 2 Report

The manuscript is a clear, concise and appropriate for this journal. The objective of the study is presented intends to show the improvement of electron transport layer(ETL) in PSC devices. It also makes a valuable contribution to the renewable energy fields.

Author Response

Thank you very much for the comments. 

Reviewer 3 Report

The manuscript ‘MDACl2-modified SnO2 film for efficient planar perovskite solar cells’ authored by Xiao et al. proposes an effective strategy for ETL modification by doping MDACl2 in the SnO2 layer. The surface defects of the SnO2 film are effectively passivated which contributes to the conductivity enhancement. More importantly, the MDACl2 modified SnO2 ETL acts as a good template for the formation of high-quality perovskite film, achieving a high PCE of 22.30% in n-i-p planar device structure. This work is well designed, and the manuscript is nicely written. I thus recommend accepting this manuscript in Molecules after the authors address a few minor comments.

A couple of comments:

1.   Introduction: There is lack of performance comparison for n-i-p planar perovskite solar cells based on conventional TiO2 ETL and SnO2 ETL by referring some recent advanced reports such as DOI: 10.3390/nano10010181, DOI: 10.1021/acsaem.1c03848, and DOI: 10.1016/j.joule.2022.07.004, which should be properly cited.

2.     Experimental Section should be either before Results and Discussion or after Conclusion.

3.     Table 1: The maximum value (champion cell) for each parameter should be highlighted.

4.     In Figure 1b-e, the number of tested devices should be noted.

5.     Figure S2c: The way in Tauc plot to extract the band gap is not accurate. The first kink point should be considered for the extraction, which is around 3.85 eV. Please rectify.

6.     The integrated Jsc (24.06 mA cm-2) measured in EQE data is less than the measured Jsc (24.65 mA cm-2) in J-V curve for the case of the best SnO2-MDACl2-based device, which does not make sense. Can the authors comment on this?

7.     Figure S1: The roughness around 8 nm is quite high compared to the conventional TiO2 compact layer. What is the thickness of the actual SnO2 layer in the device? This information should be noted.

8.     The lifetime unit (ns) should be highlighted in Table S3.

9.     It is noted that the CB and VB levels of triple-cation perovskite in this work are -3.89 and -5.46 eV, which are much higher than commonly reported values, e.g., -4.05 eV for CB and -5.65 eV for VB. How did the authors obtain these values? Please show either the reference or experimental data.

10.  Page 4 Line 127: ‘reduced’ should be ‘increased’.

Reviewer 4 Report

Please check the attached word file

Round 2

Reviewer 4 Report

The authors have addressed the comments and the manuscript have been improved. Some minor clarifications are needed.

In the cover letter addressing the comments, the authors note that the XPS peak around 380 eV is attributed to K2s. If this the case, then the K2p peak must be present at around 292 eV with about double intensity compared to K2s. The authors should show this. Indeed, K+ can be used as stabilizer in the commercial colloidal SnO2 and it is worth to inform the readers.

Finally, the authors show in lines 348-354 that a more hydrophobic substrate is beneficial for the growth of the perovskite layer. Some reports show the opposite effect. However, the authors can and the following work that shows that a hydrophobic substrate is required for the growth of a good perovskite layer. https://doi.org/10.1039/D1TC02726C
